# Exploring Health Behaviours, Attitudes and Beliefs of Women and Men during the Preconception and Interconception Periods: A Cross-Sectional Study of Adults on the Island of Ireland

**DOI:** 10.3390/nu15173832

**Published:** 2023-09-01

**Authors:** Emma H. Cassinelli, Abby McClure, Ben Cairns, Sally Griffin, Janette Walton, Michelle C. McKinley, Jayne V. Woodside, Laura McGowan

**Affiliations:** 1Centre for Public Health, School of Medicine, Dentistry and Biomedical Sciences, Queen’s University Belfast, Belfast BT12 6BA, UK; 2Institute for Global Food Security, School of Biological Sciences, Queen’s University Belfast, Belfast BT9 5DL, UK; 3Department of Biological Sciences, Munster Technological University, T12 P928 Cork, Ireland

**Keywords:** preconception health, reproductive-aged adults, young adults, health behaviour, diet, physical activity, stress

## Abstract

Preconception health is increasingly seen as a key target for improving population health in the UK and Ireland, yet little is known about the attitudes and beliefs of adults regarding preconception care strategies. This online cross-sectional survey aimed to explore the health behaviours, attitudes and beliefs of adults of reproductive age in regard to preconception health and care. The survey was developed by reviewing the previous literature and engaging with those from the target group. It is one of the first surveys to assess the attitudes and beliefs of adults of reproductive age across the Island of Ireland regarding preconception health and care. Results from 386 individuals with a mean age of 29.9 ± 10.3 years were included. A variety of health behaviours, attitudes and beliefs were investigated, with differences being identified between women and men and between participants with or without children (i.e., in the preconception or interconception stage). The majority of respondents held beliefs that preconception care was important, but there was greater emphasis on women than men in terms of the need to engage in health-promoting preconception health behaviours. This study highlights the need to improve preconception health awareness in women and men in the preconception and interconception stage. Findings indicate that efforts to improve preparation for pregnancy among adults of childbearing age are needed, to ensure optimal engagement in preconception health behaviours, with efforts being tailored based on sex and parental status.

## 1. Introduction

Preconception health is a broad term that refers to the overall health of non-pregnant individuals of reproductive age [1,2]. According to the National Institute for Health and Care Excellence (NICE), reproductive age for females ranges from 16 to 45 years old [3]. However, reproductive capabilities and fertility in males are often sustained for longer, with the literature suggesting that the volume of male sexual organs starts to decline after 60 years of age [4], and hence, fertility may become compromised. Approximately 64% and 65% of the population in Northern Ireland (NI) and Ireland, respectively, fall within the age range of 15–64 years old [5,6], suggesting that reproductive-aged individuals represent a significant proportion of the population.

Preconception care includes a range of preventative strategies that aim to support the adoption of healthy behaviours across the reproductive life-course, to improve fertility and birth outcomes should a pregnancy occur, and reduce health risk factors and non-communicable diseases in the offspring [7]. It can also enhance overall health even if a pregnancy is not achieved [7]. Many behavioural, biomedical, demographic, social and environmental factors can affect preconception health, including micronutrient supplementation (e.g., folic acid), social support, weight status, parental age, alcohol consumption, smoking, medication use and domestic abuse [8]. In studies researching preconception behaviours, folic acid supplementation, alcohol consumption and smoking have consistently been topics of focus, predominantly in women [7]. However, the wider variety of preconception health attitudes and behaviours is rarely explored [9]. In terms of biomedical factors, a greater perceived risk of pregnancy-related complications introduced, for example, by the presence of chronic medical conditions, may also affect the adoption of preconception health-promoting behaviours [10]. Further research is, however, needed on this subject [10]. Social and environmental factors, including social norms and personal characteristics such as ethnicity, economic factors (e.g., income, insurance coverage) and geographic location may further influence individuals’ preconception health attitudes and behaviours [2,11].

Having children can also influence engagement in health-promoting behaviours [12], and interconception care (i.e., care provided between the end of a pregnancy and the start of the next pregnancy) has been shown to improve parental long-term health [13]. Interconception health can address risk factors present in a previous pregnancy and optimise birth outcomes such as preterm birth, low birthweight infant or foetal loss [13]. Additionally, interventions among women with young children have been successful at increasing folic acid intake [14,15], and behavioural interventions (e.g., physical activity programmes, dietary interventions) have been shown to be effective in terms of weight loss among post-partum women [16]. This is of interest given the additional risks presented by excessive weight gain between pregnancies on subsequent pregnancies, including an increased risk of gestational diabetes mellitus, caesarean section and large for gestational age infants [17].

Research suggests that most women recognise the importance of the preconception stage [9,18], especially regarding physical activity, diet and micronutrient supplementation, and they may implement certain behaviour changes, including taking pre-pregnancy supplements, engaging in physical activity, following an adequate diet and limiting their alcohol consumption [9], although population rates remain suboptimal [19]. Male partners often make little to no behavioural changes (e.g., reducing alcohol consumption, improving diet) [9,20], perhaps due to a lack of perceived need [9] or limited knowledge of the paternal factors that can negatively affect foetal and infant health [21]. This may be considered unsurprising because, until recently, women have often been the sole target of preconception health optimisation strategies. However, the health profiles of men should not be overlooked, given that their behaviours can also impact pregnancy and child outcomes [22]. While the interest in including men in research addressing the preconception stage is increasing, there is still a need to raise awareness about paternal preconception health [20] and to include men in studies specifically exploring preconception health attitudes and behaviours [7].

The limited awareness of modifiable preconception health and care practices among individuals of reproductive age (particularly males) [23] and the fact that approximately 45% of pregnancies in Britain are recorded as unplanned or ambivalent [24], suggest that strategies implemented across the reproductive years to optimise health behaviours are needed to have an impact on preconception health. In this study, a cross-sectional survey was developed to examine the self-reported health status and pattern of health behaviours of women and men aged 18–60 years old across the Island of Ireland, and to explore their general attitudes towards preconception health and care and their beliefs regarding the importance of preconception health and care practices and behaviours. In this study, the preconception stage was defined as at least three months prior to conception; research has shown that the two to three months prior to conception are critical to optimising gamete function and placental development in the early stages for individuals who eventually achieve conception [25].

## 2. Materials and Methods

This was a quantitative, cross-sectional study, which combined data from two questionnaire-based surveys conducted in NI in 2018 and in Ireland in 2020 (all data was collected prior to the COVID-19 outbreak in early 2020). The questionnaire was developed via an iterative process (described below) and distributed online using Qualtrics (https://www.qualtrics.com/uk/ accessed on 01 December 2017). The questionnaire was distributed opportunistically to relevant networks including university staff and students via email circulars and newsletters, sports teams, relevant organisations (e.g., Family Planning Association) and widely on social media (i.e., Facebook, Twitter, WhatsApp).

A sample size calculation was broadly based on approximately 65% of the population in NI falling within the reproductive years. Using the Qualtrics online survey sample size calculator tool, it was estimated that with 95% confidence level, a population size of approximately 1,235,000 (65% of NI population overall) and a 5% margin of error, a sample size of *n* = 385 was needed. However, given that little is known on this topic, there were a number of exploratory hypotheses which were investigated using two-tailed significance with the appropriate parametric or non-parametric tests, depending on the distribution of the data, which was checked for all analyses.

### 2.1. Participants and Ethics

To be eligible for this survey, respondents had to be adults aged between 18 and 60 years. The survey included both women and men across the UK and Ireland, with and without children. There were no other inclusion/exclusion criteria for participation (details on how the study was presented to participants is shown in Appendix A).

Ethical approval was obtained in Northern Ireland by the Queen’s University Belfast School of Medicine, Dentistry and Biomedical Sciences Research Ethics Committee (reference code: 17.50v2, approval date: 29 November 2017) and in Ireland by the Cork Institute of Technology (now Munster Technological University) Ethics Committee (reference code: V1, approval date: 24 February 2020).

### 2.2. Questionnaire Development

The questionnaire was developed following a review of the literature with validated measures used where possible and any new measures being subject to Personal and Public Involvement (PPI, the preferred term used in NI instead of Patient and Public Involvement and Engagement) activities. These activities were carried out with the target demographic group (e.g., adults of reproductive age) and comprised discussions on the general topics, including question sensitivities, framing and wording, response categories and any other feedback. This was followed by pilot testing and changes were implemented accordingly.

The questionnaire covered various topics, including pregnancy planning, attitudes towards preconception health and care, beliefs on preconception health and care, sources of preconception health information, health status and current health behaviours. This study specifically reports on the health status and health behaviours, beliefs on preconception health and care, and attitudes towards preconception health and care. The full questionnaire is included in Appendix A.

### 2.3. Measures

One question in the survey, which was first carried out in NI and then in Ireland, was slightly adapted to better suit the respective context. The question pertained to the description of a standard drink. In line with suggested drinking units, ‘one drink’ was changed from ‘one regular glass of wine, one measure of spirits or one half pint of ordinary strength beer’ to ‘one small glass of wine, one pub measure of spirits or one half pint of beer/stout/lager (ordinary strength)’ [26]. Other measures are detailed below.

#### 2.3.1. Demographic Characteristics and Health Status

Basic demographic information was obtained including sex, age, parental status (i.e., having children or not), ethnicity, country of residence and socio-economic and educational status, alongside self-reported Body Mass Index (BMI) (see Appendix A to view wording of questions and categories).

The questionnaire asked respondents to provide their ‘sex’, rather than their ‘gender’ or ‘gender identity’. This was to indicate the reproductive category they belonged to, rather than participants’ societal role or inner sense of self, as sex is pertinent in discourse relating to reproduction given the biological implications [27]. The terms woman/women and man/men were used in the present study, but evidence generated may also apply to people holding a different gender identity than that discerned at birth. Parental status was recorded to differentiate respondents who had no children (considered in this paper as in a preconception stage) and those with children (considered in an interconception stage, although it is recognised that some of the people in this group may have already completed their families). Respondents were considered in the interconception stage if they reported having children, whether biological or otherwise, irrespective of whether they reported being pregnant at the point of survey completion (or their partner). Socio-economic status was categorised using the highest household income earner’s job title and the four categories of the National Statistics Socio-economic Classification (NS-SEC) system [28]. A further category was added to account for students in full-time education. A 13-point scale ranging from no formal education to doctoral level was used to assess educational status and included vocational-type training. This was then condensed into four groups, namely ‘No qualifications or compulsory level’, ‘Secondary/Further education’, ‘University or higher’ and ‘Still in full time education’, based on the International Standard Classification of Education (ISCED) groups [29]. Moreover, BMI was calculated from self-reported height (in metres) and weight (in kilograms) and categorised as ‘underweight’ (<18.49 kg/m^2^), ‘healthy weight’ (18.5–24.99 kg/m^2^), ‘overweight’ (25.0–29.99 kg/m^2^) and ‘living with obesity’ (≥30 kg/m^2^) [30]. Respondents also self-described their weight on a five-point scale ranging from ‘very underweight’ to ‘very overweight.’

#### 2.3.2. Current Health Behaviours

Dietary quality was assessed using the ‘Eating Choices Index’ (ECI), with a higher score indicating consumption of a higher-quality diet [31]. Physical activity levels were measured using the single-item measure ‘Physical Activity Assessment for Public Health’ [32]. The questionnaire also covered additional health-related behaviours, including supplement use, smoking, alcohol, sleeping habits and stress levels. These items in the questionnaire were based on available evidence. Examples of questions are ‘Do you consider yourself to be under continual stress?’ and ‘Do you consider your current sleep: Poor, Average, Good, or Very good?’ (see Appendix A for question wording and categorisation) [3,33,34].

#### 2.3.3. Attitudes towards Preconception Health and Care

Eleven items on general attitudes towards preconception care were included, following a Likert Scale format ranging from ‘Strongly disagree’ to ‘Strongly agree’. For example, respondents were asked about their perceived importance of preconception care and its effects on pregnancy outcomes. The questionnaire also included questions on the appropriateness of sources of preconception advice (e.g., GP, practice nurse, friends, internet), adapted from the ‘Preconception care questionnaire for general practitioners, practice nurses, midwives and health visitors’ [35], and preconception health behaviours of family and friends (e.g., pregnancy planning). All items can be found in Appendix A.

#### 2.3.4. Preconception Health Beliefs

Participants were asked to rate the importance of health-promoting behaviours (e.g., no smoking, no alcohol consumption, etc.) in the preconception stage, alongside the importance of maintaining a healthy weight. These beliefs were assessed by a five-point Likert scale, ranging from ‘Strongly disagree’ to ‘Strongly agree’, and participants gave separate ratings to indicate how important they believed the behaviours to be for their sex and for the opposite sex. For example, respondents were asked to rate whether they believed stopping smoking cigarettes before conception was important for women and then separately for men, or whether reducing stress preconception was important for women and then separately for men (see Appendix A to view other questions on preconception health beliefs). These items aimed to identify any differences in sex-specific perceptions and responsibilities for preconception care and preconception health behaviours.

An additional item of the questionnaire asked participants to rank the importance of a number of preconception health-promoting behaviours. These behaviours were either adapted from the ‘Health Practices in Pregnancy Questionnaire II’ (e.g., actively reducing or managing stress), or added based on current preconception guidelines (e.g., use of e-cigarettes) [3,33,36] (see Appendix A to view the full list of behaviours included).

### 2.4. Statistical Analysis

Questionnaire results were analysed using IBM SPSS (Version 25). Data were analysed using descriptive analyses and examined according to respondents’ sex, parental status (i.e., preconception vs. interconception) and age category. Where appropriate, independent *t*-tests or the non-parametric equivalent (for continuous variables) or Chi-square tests (for categorical variables) were used for significance testing at the *p* = 0.05 level, following an assessment of distribution and normality. To visually present findings regarding attitudes and beliefs relating to preconception health and care practices, stacked bar charts were generated using Microsoft Excel (version 2307).

## 3. Results

### 3.1. Demographics

Of the *n* = 528 participants who responded to the online survey, *n* = 386 were eligible for analyses (i.e., completed ≥95% of the questionnaire). Of these, 76.9% identified as women, and 97.9% were of white ethnicity. Respondents’ age ranged between 18 and 60 years, with an overall mean age of 29.9 ± 10.3 years. The mean age for respondents without children was 24.0 ± 5.6 years, whereas for respondents with children, it was 40.4 ± 8.2 years. For the purposes of analysis, age groups were categorised as younger adults (18–27 years) and older adults (28–60 years), based on a median split. Regarding employment status, 60.5% worked in higher managerial, administrative and professional occupations, and 73.2% had either achieved an undergraduate or postgraduate university degree or were still in full-time education. The majority of respondents (62.9%) had no children and were, therefore, in the preconception stage. Only one respondent from the lower age group reported having children (*p* < 0.001). A detailed breakdown of the sample’s demographics is presented in Table 1.

### 3.2. Health Status and Current Health Behaviours

Based on respondents’ BMI, calculated from self-reported height and weight, 3.3% of respondents were classified within the underweight category, 55.5% in the healthy weight category, and 28.5% and 12.7% were classified within the overweight or living with obesity categories, respectively. Participants in the preconception stage (i.e., who did not have any children), had a lower mean BMI than those considered in the interconception stage, who reported having children (*p* < 0.001). When asked about their perceived weight, 4.4% of respondents identified as underweight, 34.7% reported having ‘about the right weight’, and 34.7% and 10.2% reported ‘slightly’ or ‘very overweight’, respectively. A significant association was found between self-reported BMI and perceived weight status when performing a Chi-square test (*p* < 0.001), overall showing a high level of correspondence between self-reported BMI and perceived weight.

The mean ECI score was 13.3 ± 2.8, denoting a relatively high dietary quality in the sample, and over half of the respondents (57.3%) reported not taking vitamin supplements. Parental status was significantly associated with smoking habits (*p* < 0.001) and frequency of alcohol consumption (*p* < 0.001). For example, while the proportion of individuals who reported never smoking was similar in both groups, respondents who were in the interconception stage were more likely to be former smokers compared to respondents in the preconception stage (23% vs. 9.2%), who instead were more likely to be current smokers (20.2% vs. 7.2%). There were similar proportions of individuals in the preconception and interconception stage that reported never consuming alcohol (13.8% and 14.9%, respectively). Compared to respondents in the preconception stage, a greater proportion of respondents in the interconception stage reported drinking regularly, either 1–2 or 3–7 times a week (34.8% vs. 26.8% and 15.6% vs. 4.6%, respectively). However, fewer respondents in the interconception stage reported only drinking ‘occasionally’ than those in the preconception stage (34.8% vs. 54.8%). Respondents in the interconception stage were less likely to meet sleeping requirements than respondents in the preconception stage (*p* = 0.005), but were also less likely to report being under continual stress (*p* = 0.046).

Sex and parental status were not significantly associated with vitamin use, physical activity and average weekly alcohol consumption (*p* > 0.05). Additional data on health-related behaviours are presented in Table 2.

### 3.3. Attitudes towards Preconception Health and Care

In terms of respondents’ attitudes towards preconception care, 80.1% of women and 79.8% of men strongly disagreed or disagreed with the statement ‘Preconception care does not have any effect on pregnancy outcome.’ Additionally, about a third of the sample stated that preconception care is a high priority for them, and half of the sample would use a dedicated clinic for preconception care, if available. Most respondents strongly agreed or agreed that preconception care is important for women (women respondents: 82.2%, men respondents: 85.2%) and, to a lesser extent, for men (women respondents: 61.3%, men respondents: 59.6%). No statistical differences were detected based on sex and parental status regarding the effect of preconception care on pregnancy outcomes, the use of a dedicated clinic for preconception care and the importance of preconception care (*p* > 0.05). There was uncertainty regarding how much evidence was available on preconception care, especially among respondents in the preconception stage (*p* = 0.011).

Most respondents strongly disagreed or disagreed with the statement ‘My friends/family do not plan for pregnancy’, with statistical differences identified between people with children compared to those without (60% vs. 41.2% reported disagreement, respectively; *p* = 0.001).

In terms of sources of information, while a larger proportion of men than women strongly agreed or agreed that their GP is an appropriate person to offer preconception advice (79.7% vs. 68.3%, *p* = 0.044), more women than men strongly agreed or agreed that friends and family (28.1% vs. 13.5%, *p* = 0.012) and the internet (41.2% vs. 25.9%, *p* = 0.004) are appropriate to gather preconception advice.

Results by sex are presented in further detail in Figure 1a,b. Results by age and parental status are shown in Appendix A.

### 3.4. Preconception Health Beliefs

Certain differences in health beliefs across subgroups of the sample were identified. For example, more women than men strongly agreed or agreed with the importance of: women taking folic acid supplements (73.3% vs. 66.3%, *p* = 0.008), men achieving a healthy weight (60% vs. 52.8%, *p* = 0.007), men being physically active (61.1% vs. 56.8%, *p* = 0.036) and men reducing stress (26.7% vs. 22.5%, *p* = 0.019) in the preconception stage. More respondents in the interconception stage, compared to those in the preconception stage, strongly agreed or agreed that women should take folic acid supplements (86.5% vs. 63.2%, *p* < 0.001). Instead, respondents in the preconception stage, compared to those in the interconception stage, were more likely to strongly agree or agree that it is important for women to get at least 7–8 h of sleep per night (66.6% vs. 56%, *p* = 0.041) and to reduce stress (84.9% vs. 79.4%, *p* = 0.047) in the preconception stage. Beliefs about smoking and alcohol consumption were consistent across groups, with differences detected only between women and men on the importance for men to stop smoking cigarettes and e-cigarettes during the preconception stage. Specifically, more women, compared to men, strongly agreed or agreed that it is important for men to stop smoking cigarettes (91.2% vs. 78.7%, *p* < 0.001) and e-cigarettes (79.3% vs. 68.2%, *p* = 0.006).

Men were more likely to strongly agree or agree than women about the importance of visiting healthcare professionals in the preconception stage to discuss health practices for both women (73.1% men vs. 52.7% women, *p* = 0.001) and for men (56.2% men vs. 38.9% women, *p* = 0.002). Those in the preconception stage were also more likely to strongly agree or agree with this statement for women (66.1% respondents without children vs. 42.6% with children, *p* < 0.001) and men (52.7% respondents with children vs. 25.5% without children, *p* < 0.001). Details on selected respondents’ health beliefs by sex are presented in Figure 2a,b. Additional results, including analyses by age and parental status, can be found in Appendix A.

When asked to rank the importance of specific health behaviours in the preconception stage, in all circumstances participants felt that smoking cessation was of highest importance (see Table 3).

Beliefs on maintaining a healthy weight and diet were also ranked consistently highly by women and men, although men ranked being physically active more highly than women. In general, women placed greater importance on addressing mental health issues in the preconception stage than men, and this view was ranked of higher importance by those in the preconception stage, compared to those in the interconception stage (see Appendix A). Both women and men ranked cutting out alcohol and stopping the use of e-cigarettes during the preconception stage lower down the list. Obtaining an adequate amount of sleep and taking vitamin D were ranked of least importance.

Although solely asked to female participants, the use of folic acid during the preconception stage was ranked quite highly overall (ranked third). However, women in the preconception stage and younger women (18–27 years old) ranked this behaviour lower down the list (i.e., less important). Rankings of preconception behaviours by women for women and for men based on age and parental status are presented in Appendix A.

## 4. Discussion

A cross-sectional survey was conducted with the aim to explore self-reported health status, patterns of key health behaviours, general attitudes towards and beliefs on preconception health and care. The study reported on responses from *n* = 386 women and men aged 18–60 years old across the Island of Ireland. There were *n* = 239 (62.9%) respondents who reported being in the preconception stage and *n* = 141 (37.1%) in the interconception stage.

There were differences in responses given based on whether respondents reported as being women or men, or in the preconception or interconception stage. Findings overall suggested that efforts are needed to increase the awareness of preconception health in reproductive-aged adults.

### 4.1. Health Status and Current Health Behaviours

The current health status and behaviours of adults of reproductive age are of significance, especially given the impact that parental health can have on potential offspring. There were differences identified based on which behaviours respondents in the preconception versus interconception stage engaged in, suggesting that having children and transitioning to parenthood can lead to changes in adult health-related behaviours. In this study, respondents with children were more likely to be former smokers compared to respondents without children who, instead, were more likely to be current smokers. More respondents with children than without children reported drinking regularly (i.e., 1–2 or 3–7 times/week). However, this finding solely reports on the frequency of alcohol consumption rather than the quantity. Indeed, the question ‘How often do you normally drink alcohol?’ yielded different results than the question ‘How many alcoholic drinks, on average, would you consume each week?’, which did not differ significantly according to parental status.

Respondents in the interconception stage were less likely to meet sleeping requirements or report being under continual stress than respondents in the preconception stage. They also had a higher self-reported BMI than respondents in the preconception stage, a finding also reported in previous studies [37], suggesting that a better understanding of weight gain in pregnancy and after pregnancy is needed. Behavioural differences based on people’s parental status have also been reported in the literature; for example, studies have shown that women without children engage in fewer high-risk health behaviours (e.g., low physical activity, unhealthy eating habits) than women with children [12], but there is also evidence suggesting that health-compromising behaviours are less prevalent postpartum than in the preconception stage [13].

Children’s characteristics (e.g., age) and behaviours have been shown to affect parents’ behaviours [38]. While the present study did not investigate the dynamics of parent-child interactions and the potential variations in behaviours reported by individuals in the interconception stage based on their child or children’s characteristics, these may be considered when tailoring support for individuals in the interconception stage, and should be explored in future research. A previous survey has suggested that most women were aware of the importance of making healthy behaviour changes prior to pregnancy and regarded making these changes as very easy or easy [18]. However, in the UK, only about half of women planning a pregnancy report making changes in preparation for pregnancy [39], and high-risk health behaviours in women and men planning a pregnancy or actively trying to conceive are frequently reported [12]. There are factors that have been shown to influence patterns of behaviours prior to conception, for example the presence of a chronic medical condition [10], but these were not investigated in the present questionnaire. Overall, the findings from this study and the surrounding literature suggest that greater efforts to increase the public’s awareness of health behaviours and improved support to facilitate any behaviour changes should be encouraged in individuals of reproductive age. Further research on preconception behavioural intentions and how they may translate into health-promoting behaviour changes at this time would be valuable [7].

#### Comparisons with Current Preconception and/or Interconception Recommendations

The Clinical Knowledge Summary (CKS) by NICE on preconception [3] is a resource that reports advice for all women considering or planning a pregnancy. Optimal behaviours reported which are relevant to this study include the achievement of a balanced diet (e.g., following the Eatwell Guide) [40] and healthy weight, smoking cessation and alcohol avoidance. The majority of participants self-reported a healthy weight (55.5%), never smoking (70.2%) and never or occasionally drinking alcohol (61.5%). In terms of diet, the mean ECI score was 13.3 ± 2.8, suggesting that the sample had a relatively high diet quality. Based on these findings, it appears that the behaviours reported by participants in this study were partly aligned with the advice provided in the NICE CKS resource; however, greater support is still needed. It should also be noted that there are other behaviours of critical importance discussed in the NICE resource, including addressing the use of illicit or recreational drugs, risks from exposure to hazardous substances or radiation, medication use, cervical screening and immunisations, that were not specifically included in the questionnaire. Chivers and colleagues compared health behaviours of women planning a pregnancy with national recommendations in Australia and found that women were failing to adhere to recommendations [41]. They too highlighted the need to optimise women’s behaviours to improve alignment with current preconception recommendations [41]. This is also consistent with findings by Inskip and colleagues, revealing that only a small proportion of women in the UK who plan on becoming pregnant are likely to engage in health-promoting behaviours and adhere to recommendations [42].

### 4.2. Attitudes towards Preconception Health and Care

#### 4.2.1. Importance of Preconception Care

Most of the respondents were aware of the importance of preconception care for individuals of childbearing age, especially for women, and of the impact it can have on pregnancy outcomes. While previous studies have also suggested that most women recognise the importance of the preconception stage [9], findings from focus groups conducted in NI have shown a lack of in-depth awareness surrounding the importance of preconception health, beyond a general understanding that health optimisation at this stage would be beneficial [21]. Although not detected in the present study, past research has also suggested that women with no previous experience of pregnancy may be more likely to regard preconception care as essential than women who have experienced pregnancy [43]. Therefore, targeted approaches (e.g., based on sex and parental status) should be encouraged to promote awareness and help translate awareness into health-promoting behaviours [43] across the reproductive spectrum.

#### 4.2.2. Pregnancy Planning

A further attitude explored in this study was pregnancy planning. Just under half of the respondents suggested that their friends and family plan for their pregnancies, which is consistent with recorded pregnancy planning levels in Britain [24]. Studies in Ireland have, instead, suggested that >30% of pregnancies were unplanned [44]. Analyses of routinely collected maternity data in NI reported a higher proportion of planned pregnancies (72.4%) [45], but these data may be influenced by bias due to how they were collected.

Findings suggest that efforts to encourage pregnancy planning and preparation for pregnancy and parenthood for modifiable health-related behaviours should be explored, with the aim of improving family building experiences and allowing health optimisation prior to conception for all. This may be especially relevant for people with pre-existing medical conditions.

#### 4.2.3. Sources of Information

Approximately half of the respondents reported they would potentially use a dedicated clinic for preconception care, if available. Seeing a GP, a practice nurse or a community midwife was deemed the most appropriate means to receive preconception advice, although this varied by respondent’s sex (e.g., more men than women strongly agreed or agreed that their GP is an appropriate person to offer preconception advice, more women than men strongly agreed or agreed that friends and family and the internet are appropriate to gather preconception advice). This is consistent with other studies’ findings, showing that healthcare professionals not only were trusted by the public [18,46] but also that healthcare professionals themselves considered preconception care to be best delivered by them [42]. However, issues with the accessibility of healthcare professionals have previously been reported [46], especially in light of the COVID-19 outbreak, and less than half of women planning a pregnancy seek medical or health advice to prepare for conception [41]. Additionally, people may regard visiting GPs for preconception advice as ‘wasting an appointment’ [21].

More women than men stated a preference for receiving preconception advice from family and friends and from the internet, while more men than women preferred GPs. Comparative analyses of people’s trust in sources of fertility-related information versus their usage have suggested that family and friends, as well as general online searches, are used more than they are trusted [46]. Previous research has also shown that individuals trust governmental or medical websites more than non-medical ones [9,46]. The internet is often cited as a source of information [21,47], with issues of misinformation due to poor quality or usability of advice, misunderstandings and overabundance of information still being reported [48,49]. The present study was conducted before the COVID-19 pandemic, thus, prior to a greater dependence on technology for the delivery of healthcare. Results may, therefore, reflect people’s preferences for preconception health advice before the pandemic and before healthcare services had to adapt and change to rely more on technology. Given the current frequent use of the internet coupled with the great trust placed on healthcare professionals, public health messaging using technology (e.g., social media, health apps) driven or endorsed by healthcare professionals or medical bodies could aid the promotion of preconception health amongst people of childbearing age [18]. The #ReadyforPregnancy campaign in England is an example of a wide range of professionals successfully collaborating on a digital initiative to provide advice on preparing for pregnancy [50].

It is important that information needs are met through the provision of reliable and trustworthy advice that can aid preconception health optimisation, and findings from this study suggest using a range of avenues to reach a wider section of the population (e.g., using different approaches for men versus women). Based on findings from a cross-sectional survey in Australia, a checklist was suggested by women as the most useful resource to be used both before conception and during pregnancy to retrieve information in a novel way [18]. The repeated delivery of information from different sources has also been proposed in the literature, with an emphasis on avoiding the addition of unnecessary pressure or stress on adults of reproductive age [51].

### 4.3. Preconception Health Beliefs

Certain sex-based differences were detected regarding behavioural factors, especially related to preconception health-optimising behaviours for men (i.e., the importance for men to be physically active, achieve a healthy weight and reduce stress). Overall, these results are in line with the previous literature, which indicated that beliefs that women should perform preconception health behaviours were stronger than beliefs about what men should do [11]. Additionally, these findings are reflected in research suggesting that women’s partners, perhaps due to limited awareness or perceived need, often fail to achieve behavioural changes prior to conception [9]. This may suggest that there is still a need to investigate the effects of men’s behaviours on pregnancy and infant outcomes, and that greater paternal awareness should be pursued [52]. Despite the differences in beliefs held between women and men, when asked to rank the most important preconception behaviours, smoking avoidance, consumption of a healthy diet and achievement of a healthy weight were considered highly important from, and for, both women and men. Taking a preconception folic acid supplement, important to reduce the risk of neural tube defects [3], was also generally ranked highly by women, although data revealed there may be informational gaps, especially for women in the preconception stage and younger women. This lack of awareness is concerning, especially given that the study sample was relatively well educated, and may highlight a need for public awareness initiatives on this topic, specifically targeted at women in the preconception stage and younger women. Indeed, having children changed how women ranked the importance of preconception folic acid supplementation.

Analyses of maternity data in England have shown that most women do not take folic acid supplements before pregnancy [19], but care provided in the interconception stage could increase women’s folic acid intake [53]. The recent decision to supplement flour with folic acid in the UK may also represent a step towards better folic acid supplementation at a population level, once the policy is fully and effectively implemented.

Overall, robust research is needed to elucidate the association between preconception beliefs and outcomes during pregnancy, also considering the wider complex systems influencing health.

### 4.4. Strengths and Limitations

This study included the development of an in-depth questionnaire that adopted validated measures, a wide variety of literature and PPI collaboration. The questionnaire assessed an extensive range of health behaviours, and the stratification by sex and parental status (i.e., being in the preconception or interconception stage) enabled a novel understanding of preconception health and care beliefs and attitudes within the Island of Ireland, which have not yet been extensively explored. However, the distinction made between respondents in the preconception and interconception stages based on whether they reported having children or not was not indicative of respondents who were expecting a child at the time of survey completion. Although this was only the case for a small fraction of participants (*n* = 32, 8.3%), this factor could have influenced responses and should be considered in more detail in future research. Similarly, while respondents’ pregnancy intention was not reported in the current study, it may also be a factor that might have influenced people’s responses.

Because the questionnaire was circulated online, findings may be influenced by sampling and selection bias. The questionnaire was circulated in NI and Ireland in separate years, namely 2018 and 2020, which may have impacted the results. However, the current study did not aim to examine differences between the two countries and health systems and instead focused on analysing the entire Island of Ireland as a whole. The fact that only 386 out of 528 respondents were eligible for analysis may be considered a limitation because it reduces the sample size, thus decreasing statistical power.

There was a greater number of women than men respondents in NI, whereas more men than women responded in Ireland. The majority of respondents were of white ethnicity, suggesting a limited transferability and generalisability of the results to other countries, although reflective of the overall population in both NI and Ireland (96.55% and 92.4% white ethnicity, respectively) [54,55]. Given the recruitment methods for this research, population groups with lower educational attainment and from lower socio-economic groupings were underrepresented in the sample, which is not unusual in questionnaire-based studies. This may be of interest as many risk factors (e.g., alcohol consumption) have been shown to differ based on socio-economic status [56] and should be explored explicitly in future research. Moreover, the assessment of participants’ grade of employment was a UK-based tool and, therefore, it may not be as applicable to responses from Ireland. Although the intention was to reach a diverse population, only one respondent from the lower age group (18–27 years old) already had children. This was not surprising as the average age of mothers at birth of the first child is above 27 years, both in NI and Ireland (29.2 years and 31.1 years, respectively) [5,57]. It is also important to acknowledge that, although the cohort aimed to include adults of reproductive age, some may not have reproductive capacity. Due to the structure of the questionnaire, it was also not possible to ascertain if diversity was achieved in terms of, for example, disability status or belonging to the LGBTQIA+ community, as sexual orientation was not asked. Additionally, only one respondent out of the full sample selected ‘Other’ in response to Sex, and so for statistical reasons, it was not feasible to include in the present analyses. A further limitation was the reliance on self-reported measures, making the findings susceptible to the influence of recall bias. For example, self-reported BMI measures were used, which research has suggested to be on average >0.9 kg lower compared to measured weight and so may have underestimated true BMI levels [58] and the potential impact of excess weight in adults.

## 5. Conclusions

A cross-sectional study was conducted in the Island of Ireland to assess the health status and health behaviours of women and men of childbearing age, alongside their attitudes towards and beliefs of preconception health and care. Differences were identified between people with and without children (i.e., in the preconception and interconception stage) and between women and men. Results suggested, for example, that greater paternal preconception health awareness is warranted, and that information needs for women in the preconception stage (e.g., about the importance of folic acid supplementation) are not being adequately met. Findings of this study highlight that effective efforts to improve awareness and support in the preconception, post-partum and interconception stage are still required to optimise health behaviours, thus suggesting that future work should be undertaken to address these gaps in knowledge.

## Figures and Tables

**Figure 1 nutrients-15-03832-f001:**
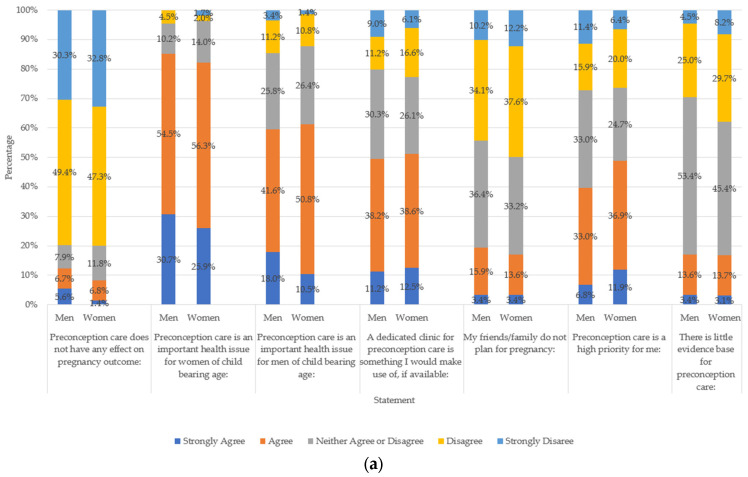
(**a**,**b**) attitudes towards preconception health and care shown by women and men respondents separately, including same-sex and opposite-sex attitudes. Chi-square tests used to explore between-group associations. *p* values shown in figures when *p* < 0.05.

**Figure 2 nutrients-15-03832-f002:**
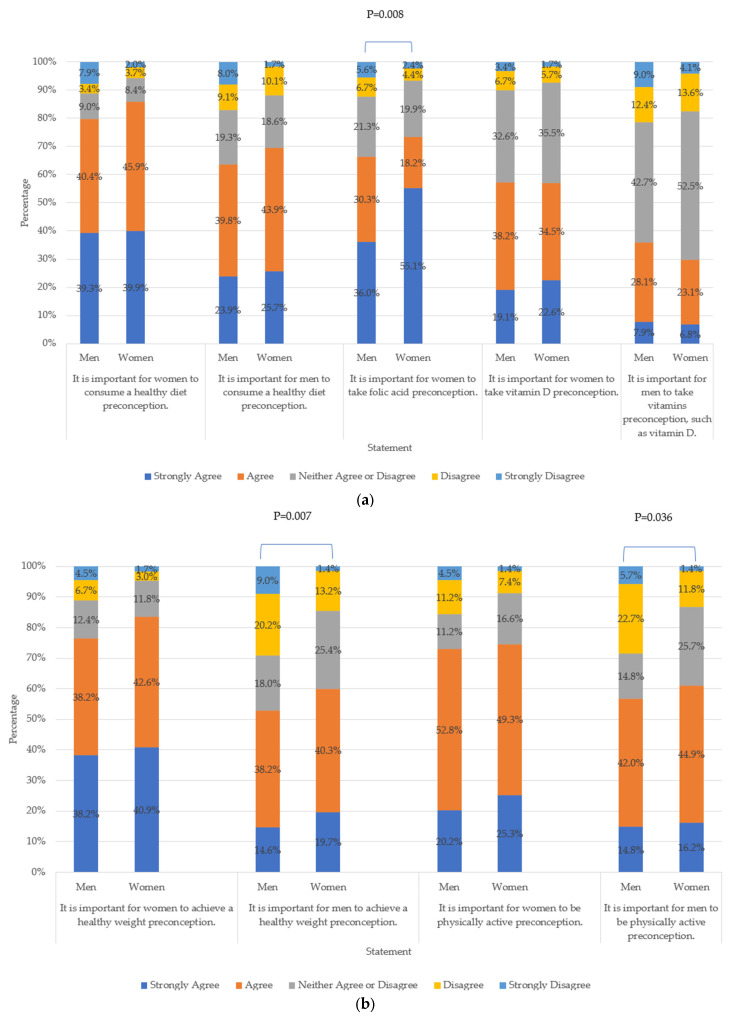
(**a**,**b**) same-sex and opposite-sex preconception health beliefs. Chi-square tests used to explore between-group associations. P values shown in figures when *p* < 0.05.

**Table 1 nutrients-15-03832-t001:** General demographic breakdown of sample.

Variable	Mean(SD)/*n*(%) All	Mean(SD)/ *n*(Column %) Men	Mean(SD)/ *n*(Column %) Women	*p*-Value	Mean(SD)/ *n*(Column %) with Children	Mean(SD)/ *n*(Column %) without Children	*p*-Value
Sex (*n* = 385) ^c^							
Men (*n* = 89)	89 (23.1%)	—	—		34 (24.1%) ^c^	52 (21.8%) ^c^	0.613
Women (*n* = 296)	296 (76.9%)	—	—		107 (75.9%) ^c^	187 (78.2%) ^c^	
Parental Status (*n* = 380)							
With Children (*n* = 141)	141 (37.1%)	34 (39.5%)	107 (36.4%)	0.613	—	—	
Without Children (*n* = 239)	239 (62.9%)	52 (60.5%)	187 (63.6%)		—	—	
Age (*n* = 333)							
Overall age in years (*n* = 333)	29.9 (10.3)	32.8 (13.1)	29.2 (9.3)	0.029	40.4 (8.2)	24.0 (5.6)	0.029
18–27 years (*n* = 167)	167 (50.2%)	39 (53.4%)	128 (49.2%)	0.617	1 (0.8%) ^c^	164 (78.1%) ^c^	<0.001
28–60 years (*n* = 166)	166 (49.8%)	34 (46.6%)	132 (50.8%)		119 (99.2%) ^c^	46 (21.9%) ^c^	
Country of Residence (*n* = 385)							
United Kingdom (incl. Northern Ireland) (*n* = 300)	300 (77.9%)	36 (40.4%)	264 (89.2%)	<0.001	102 (72.3%) ^c^	196 (82%) ^c^	0.086
Republic of Ireland (*n* = 68)	68 (17.7%)	49 (55.1%)	19 (6.4%)		31 (22.0%) ^c^	34 (14.2%) ^c^	
Other (*n* = 17) ^a^	17 (4.4%)	4 (4.5%)	13 (4.4%)		8 (5.7%)	9 (3.8%)	
Education status (*n* = 385)							
No qualifications or compulsory level (*n* = 11)	11 (2.9%)	3 (3.4%)	8 (2.7%)	0.091	6 (4.3%)	5 (2.1%)	<0.001
Secondary/ further education (e.g., NVQ) (*n* = 92)	92 (23.9%)	30 (33.7%)	62 (20.9%)		31 (22%) ^c^	59 (24.7%) ^c^	
University or higher (UG or PG degree) (*n* = 205)	205 (53.2%)	41 (46.1%)	164 (55.4%)		102 (72.3%) ^c^	100 (41.8%) ^c^	
Still in full-time education (*n* = 77)	77 (20%)	15 (16.9%)	62 (20.9%)		2 (1.4%)	75 (31.4%)	
Socio-economic class (*n* = 309) ^c^							
Higher managerial, administrative and professional occupations (*n* = 187)	187 (60.5%)	46 (66.7%)	141 (58.8%)	0.426	84 (68.9%)	100 (54.9%)	0.008
Intermediate occupations (*n* = 77)	77 (24.9%)	13 (18.8%)	64 (26.7%)		30 (24.6%) ^c^	45 (24.7%) ^c^	
Routine and manual occupations (*n* = 28) ^b^	28 (9.1%)	8 (11.6%)	20 (8.3%)		7 (5.7%)	21 (11.5%)	
Not working (*n* = 17) ^d^	17 (5.5%)	2 (2.8%)	15 (6.3%)		1 (0.8%)	16 (8.8%)	

^a^ This is indicative of ‘Others’, including the USA, Canada, Finland and the Czech Republic. ‘Other’ respondents were retained for analysis given they may have been from or resident in the UK or Ireland. ^b^ Where job description was not specified, respondents were classified as belonging to the lower employment banding, according to National Statistics Socio-economic Classification (NS-SEC) groupings. ^c^ Where ‘^c^’ is denoted, the total is inclusive of those who answered ‘Other’ (e.g., in the question regarding sex *n* = 1). This ‘Other’ response was treated as Missing in subsequent analyses. ^d^ This is a combination of ‘Never worked and long-term unemployment’ (*n* = 5) and ‘Full-time students’ (*n* = 12). Abbreviations: SD: Standard deviation, NVQ: National Vocational Qualification, UG: Undergraduate, PG: Postgraduate Independent *t*-tests and Chi-square statistics were used to determine differences and associations between groups, as appropriate.

**Table 2 nutrients-15-03832-t002:** Self-reported weight status and health behaviours breakdown of combined sample.

Variable	Overall Mean (SD)/*n*(Column %)	Mean (SD)/*n*(Column %) Males	Mean (SD)/*n*(Column %) Females	*p*-Value	Mean (SD)/ *n*(Column %) with Children	Mean (SD)/ *n*(Column %) without Children	*p*-Value
Self-reported BMI (*n* = 362) ^a^							
Overall (*n* = 362)	24.7 (4.5)	25.3 (3.9)	24.6 (4.7)	0.167	26.2 (4.5)	23.9 (4.4)	<0.001
Underweight <18.49 (*n* = 12)	12 (3.3%)	2 (2.3%)	10 (3.6%)	0.011	-	12 (5.3%)	<0.001
Healthy weight 18.5–24.99 (*n* = 201)	201 (55.5%)	39 (44.8%)	162 (58.9%)		59 (45.0%) ^d^	138 (61.1%) ^d^	
Overweight 25.0–29.99 (*n* = 103)	103 (28.5%)	37 (42.5%)	66 (24.0%)		48 (36.6%)	55 (24.3%)	
Living with obesity ≥30 (*n* = 46)	46 (12.7%)	9 (10.3%)	37 (13.5%)		24 (18.3%) ^d^	21 (9.3%) ^d^	
Self-perceived weight (*n* = 383)							
Underweight (*n* = 17)	17 (4.4%)	4 (4.5%)	13 (4.4%)	0.156	3 (2.1%)	14 (5.9%)	<0.001
About the right weight (*n* = 194)	194 (50.7%)	52 (58.4%)	142 (48.3%)		49 (34.8%) ^d^	142 (59.9%) ^d^	
Slightly overweight (*n* = 133)	133 (34.7%)	29 (32.6%)	104 (35.4%)		66 (46.8%) ^d^	65 (27.4%) ^d^	
Very overweight (*n* = 39)	39 (10.2%)	4 (4.5%)	35 (11.9%)		23 (16.3%)	16 (6.8%)	
Vitamin and mineral supplement use (*n* = 384)							
Yes (*n* = 164)	164 (42.7%)	30 (33.7%)	134 (45.4%)	0.066	59 (42.1%) ^d^	102 (42.7%) ^d^	1.000
No (*n* = 220)	220 (57.3%)	59 (66.3%)	161 (54.6%)		81 (57.9%) ^d^	137 (57.3%) ^d^	
ECI score (*n* = 382)							
Overall (*n* = 382)	13.3 (2.8)	12.9 (2.9)	13.4 (2.7)	0.127	13.7 (2.7)	13.1 (2.8)	0.037
Tertile 1 (*n* = 137) ^b^	137 (35.9%)	39 (44.8%)	98 (33.2%)	0.067	42 (29.8%) ^d^	94 (39.8%) ^d^	0.102
Tertile 2 (*n* = 107)	107 (28.0%)	17 (19.5%)	90 (30.5%)		40 (28.4%) ^d^	65 (27.5%) ^d^	
Tertile 3 (*n* = 138)	138 (36.1%)	31 (35.6%)	107 (36.3%)		59 (41.8%) ^d^	77 (32.6%) ^d^	
In the past week, on how many days have you done a total of 30 min or more of physical activity? (*n* = 385)							
Overall ( = 385)	3.0 (2.0)	3.3 (1.8)	2.9 (2.0)	0.116	2.7 (2.0)	3.1 (1.9)	0.082
None (*n* = 55)	55 (14.3%)	8 (9.0%)	47 (15.9%)	0.227	25 (17.7%)	30 (12.6%)	0.217
<5 (*n* = 236)	236 (61.3%)	56 (62.9%)	180 (60.8%)		87 (61.7%) ^d^	145 (60.7%) ^d^	
≥5 (*n* = 121)	94 (24.4%)	25 (28.1%)	69 (23.3%)		29 (20.6%) ^d^	64 (26.8%) ^d^	
Smoking Status (*n* = 382)							
Never smoked (*n* = 268)	268 (70.2%)	53 (60.2%)	215 (73.1%)	0.063	97 (69.8%) ^d^	168 (70.6%) ^d^	<0.001
Former smoker (*n* = 55)	55 (14.4%)	16 (18.2%)	39 (13.3%)		32 (23.0%) ^d^	22 (9.2%) ^d^	
Current smoker (*n* = 59) ^c^	59 (15.4%)	19 (21.6%)	40 (13.6%)		10 (7.2%) ^d^	48 (20.2%) ^d^	
How often do you normally drink alcohol? (*n* = 385)							
Never (*n* = 54)	54 (14.0%)	5 (5.6%)	49 (16.6%)	0.031	21 (14.9%)	33 (13.8%)	<0.001
Occasionally (*n* = 183)	183 (47.5%)	41 (46.1%)	142 (48.0%)		49 (34.8%) ^d^	131 (54.8%) ^d^	
1–2 times a week (*n* = 115)	115 (29.9%)	34 (38.2%)	81 (27.4%)		49 (34.8%) ^d^	64 (26.8%) ^d^	
3–7 times a week (*n* = 33)	33 (8.6%)	9 (10.1%)	24 (8.1%)		22 (15.6%)	11 (4.6%)	
How many alcoholic drinks, on average, would you consume each week? (*n* = 385)							
None (*n* = 71)	71 (18.4%)	12 (13.5%)	59 (19.9%)	0.365	25 (17.7%) ^d^	45 (18.8%) ^d^	0.504
1–10 (*n* = 221)	221 (57.4%)	53 (59.6%)	168 (56.8%)		86 (61.0%) ^d^	132 (55.2%) ^d^	
11+ (*n* = 93)	93 (24.2%)	24 (27.0%)	69 (23.3%)		30 (21.3%) ^d^	62 (25.9%) ^d^	
How many hours a night do you sleep? (*n* = 383)							
Meeting sleep requirement (7–9 h and >9 h) (*n* = 278)	278 (72.6%)	63 (71.6%)	215 (72.9%)	0.919	88 (63.3%) ^d^	185 (77.4%) ^d^	0.005
Not meeting sleep requirement (<7 h) (*n* = 105)	105 (27.4%)	25 (28.4%)	80 (27.1%)		51 (36.7%)	54 (22.6%)	
Do you consider yourself to be under continual stress? (*n* = 382)							
Yes (*n* = 170)	170 (44.5%)	32 (36.4%)	138 (46.9%)	0.103	53 (37.6%) ^d^	115 (48.7%) ^d^	0.046
No (*n* = 212)	212 (55.5%)	56 (63.6%)	156 (53.1%)		88 (62.4%) ^d^	121 (51.3%) ^d^	

^a^ In instances where respondents provided both their height in cm and ft and in, the value reported for cm was used for analysis. Where respondents provided both their weight in kg and st and lbs, the value reported for kg was used for analysis. ^b^ ECI scoring is a continuous variable, and a higher score indicated a higher dietary quality. For sample-specific comparison, the sample was divided into upper, middle and lower thirds. Tertile 1 represents the lower third and the worst ECI scores, while tertile 3 represents the upper third and the best scores. ^c^ Respondents who use cigarettes, e-cigarettes, cigarettes and e-cigarettes or may be social smokers were classified as “current smokers” due to small numbers. ^d^ Where ‘d’ is denoted, the total is inclusive of those who answered ‘Other’ (e.g., in the question regarding sex *n* = 1). Abbreviations: SD: Standard Deviation, BMI: Body Mass Index, ECI: Eating Choices Index. Independent *t*-tests and Chi-square statistics were used to determine differences and associations between groups, as appropriate.

**Table 3 nutrients-15-03832-t003:** Ranking of preconception practices by women (for women) and by men (for men).

Preconception Practice for Women (Rated by Women) ^a^	Overall Ranking	Mean (SD)	Preconception Practice for Men (Rated by Men)	Overall Ranking	Mean (SD)
Stop smoking cigarettes	1	3.18 (2.69)	Stop smoking cigarettes	1	2.68 (2.03)
Consume a healthy diet	2	4.30 (2.18)	Consume a healthy diet	2	3.65 (2.04)
Take folic acid	3	5.43 (3.67)	Achieve a healthy weight	3	4.96 (2.24)
Achieve a healthy weight	4	5.81 (2.76)	Be physically active	4	5.83 (2.47)
Address mental health issues	5	6.29 (3.38)	Stop using e-cigarettes	5	5.88 (3.02)
Stop using e-cigarettes	6	6.45 (3.39)	Cut out alcohol	6	6.26 (3.08)
Cut out alcohol	7	6.88 (3.40)	Address mental health issues	7	6.95 (2.50)
Be physically active	8	7.2 (2.69)	Visit a healthcare professional	8	6.69 (3.85)
Actively reduce/manage stress	9	7.35 (2.87)	Actively reduce/manage stress	9	6.95 (2.50)
Visit a healthcare professional	10	7.7 (3.99)	Take vitamin D	10	7.88 (2.51)
Take vitamin D	11	8.03 (2.75)	Get adequate amounts of sleep	11	8.74 (2.20)
Get adequate amounts of sleep	12	9.38 (2.43)			

^a^ Women had an additional choice compared to men, as they were asked about taking folic acid supplements.

## Data Availability

The data presented in this study are available on request from the corresponding author.

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
