# Peer review of "Exploring Health Behaviours, Attitudes and Beliefs of Women and Men during the Preconception and Interconception Periods: A Cross-Sectional Study of Adults on the Island of Ireland"

_nutrients, 2023, doi:10.3390/nu15173832_

Round 1

Reviewer 1 Report

Thank you for the opportunity to review, not only this important manuscript, but also be part of this interesting special issue.

‘Exploring health behaviours, attitudes and beliefs of women and men during the preconception and interconception periods: A cross-sectional study of adults on the Island of Ireland’ is a novel cross sectional study which provides good evidence and justification for promotion of pre- and inter-conception health. Data were collected in 2018 and 2020 and have been analysed together yielding a large sample size.

I have listed my comments per section:

Abstract

Good overview of the study

Lines 16-19, I suggest splitting this into 2 sentences

The aim is in there but it’s not explicitly highlighted. Consider making it more obvious and adding in mention of health behaviours (e.g. this online cross sectional survey aimed to assess health behaviours, attitudes and beliefs of adults of reproductive age…)

Introduction

Well presented section with key areas mentioned and introduced.

Be wary of long sentences throughout e.g. lines 49-52

Lines 42-69 contains a lot of important background information. Although it gets a little hard to follow around the middle of this paragraph. I would consider structuring this paragraph using the behavioural, biomedical, demographic, social and environmental factors as anchors for this paragraph to guide the reader through the literature/ justifications.

Paragraph starting on line 85, I would suggest emphasising where this data comes from. There is mention of Northern Ireland specific data at the start of the introduction, so I think the context of UK wide needs to be mentioned in this paragraph.

Methods

Supplementary materials 1 (questionnaire) could be presented more efficiently. Considering other researchers may use this as a questionnaire/ base of questionnaire, I would suggest making it more user friendly layout.

Good mention and justification of sex and gender.

Was pregnancy status measured? Some participants may have been pregnant/ had pregnant partners which would influence behaviours, attitudes and beliefs. Or did ‘with children’ also mean pregnant (was this made clear to participants)? This should be mentioned with additional consideration in the discussion section.

In addition, was pregnancy planning measured atall (e.g. LMUP)? This would be a very interesting angle to look at the data and would get around the issue of whether someone has ‘completed’ their family or not. Furthermore, it could link in nicely with Barker et al.’s preconception action phases, which might explain behaviours, attitudes and beliefs at different stages. Again, this should be mentioned with additional consideration in the discussion section.  

There should be a section about participants, mainly inclusion/ exclusion criteria, how many participants were being recruited, power, ethical approval…

I feel there could there be more detail about analyses, possibly structured using the key factors mentioned in the research aim (e.g. behaviours, attitudes and beliefs). At the moment, it would be challenging for someone else to conduct this analysis given the detail provided.

Results

Good overview of the results

What is the justification for having the inclusion criteria as 95% completion? A reference or justification would be useful here. How was missing data dealt with form those who were included?

Good use of tables and figures.

Would the authors consider complementing this analysis with overall scores of attitudes and beliefs? For example, could all the scores for attitudes be combined per participant to give an overall attitude score? This could then be considered between your groups with analysis? This might give a more useful and generalisable overview of the findings. Although the existing analysis should not be removed as the nuisance per question is extremely important.

Discussion

Good use of relevant and recent literature with both research and clinical implications

Discussion header needs formatting for consistency.

The opening paragraph would benefit from a high level summary of the findings, to complement the repeated aim, before delving into what the findings mean in the wider context.

Line 335. Is there any data on unplanned/ mistimed pregnancies in NI and Ireland? This would help keep the context of the findings.  

Paragraph starting line 334. Interesting distinctions between some health behaviours for preconception and interconception. I can’t help but reflect on the complexities of the interconception group as these will contain people who have young children, older children, gown up children etc. The motivations for behaviour and choice of behaviours will vary greatly between these groups. I think the diversity within these groups needs to be considered/highlighted more are tailoring support to the needs of those within the interconception group will vary greatly.  

Reviewer 2 Report

Cassinelli et al present a cross sectional study with data examining the health behaviours and attitudes of Irish men and women in the pre-conception period. Timely research, fantastic to see focus on the paternal as well as maternal diet. Overall, this is a well written and well presented study that will be of interest to readers of this journal.

Methods:

The population appears to be from a relatively affluent and healthy demographic (university educated, involved in sports. This appears to be a major limitation as these populations may not necessarily be reflective of the population as a whole.

An additional limitation is the timing of the survey between the 2 locations. COVID is likely to have had an effect on many behaviours. Were questions asked in relation to whether the individuals had been infected with COVID or whether they had an adverse long term effects?

Fantastic to see PPI involved in the design of this research. Also great to see the distinction between sex and gender as person centred language in relation to overweight/obesity.
